# OpenReview forum: "Disagreements in Reasoning: How a Model's Thinking Process Dictates Persuasion in Multi-Agent Systems"
_ICLR.cc/2026/Conference — ICLR 2026 Conference Withdrawn Submission_

### Official Review · Reviewer_63XN · 2025-10-31

**Soundness:** 3
**Presentation:** 3
**Contribution:** 2
**Rating:** 4
**Confidence:** 4

**Summary:**

The authors propose that a model's internal reasoning process is the key determinant of both its ability to persuade others and its resistance to being persuaded. Through experiments on objective (MMLU) and subjective (PersuasionBench, Perspectrum) datasets across 10 model configurations, they identify the "Persuasion Duality" - a fundamental trade-off where explicit reasoning simultaneously increases persuasive capability and resistance to persuasion. The paper also proposes an attention-based explanation for why models are easily misled and introduces a prompt-level mitigation strategy.

**Strengths:**

- The focus on cognitive architecture rather than model scale is refreshing, especially given the rise of reasoning models.


-  The paper evaluates a diverse range of models (both open and closed-source) across multiple datasets and experimental conditions, enhancing generalizability.

- The attention analysis in Section 4.1 provides valuable insights into why models are susceptible to persuasion, showing that models prioritize confident assertions over logical reasoning.

**Weaknesses:**

The paper primarily studies single-turn persuasion. Multi-turn debates might reveal different dynamics.


While the framing is nice, the core finding (reasoning makes models both more persuasive when sharing thinking AND more resistant when being persuaded) seems intuitive. The conceptual contribution would be stronger with:

- Theoretical grounding (e.g., connections to cognitive science, argumentation theory)
- Quantitative characterization of the trade-off (e.g., is it linear? Are there optimal points?)

**Questions:**

The attention analysis suggests models focus on confident assertions rather than logical content. Have you tested whether intervening on attention (e.g., masking high-attention tokens) changes persuasion outcomes? This would strengthen the causal claim.


In Section 3.3.2, you show that "mismatched" thinking content is detrimental. But what makes thinking content "high quality"? Can you characterize the features of persuasive vs non-persuasive thinking content?


Why do some models (e.g., Gemini, Qwen, Hunyuan) show inconsistent effects when enabling thinking mode as persuaders (Section 3.2)? Is this due to training differences, thinking quality, or something else?

---

### Official Review · Reviewer_CKnX · 2025-10-31

**Soundness:** 1
**Presentation:** 1
**Contribution:** 1
**Rating:** 2
**Confidence:** 5

**Summary:**

This paper shows that in multi-agent systems of LLMs and reasoning models, how a model thinks matters more than how big it is when it comes to persuasion. Through large-scale experiments, the authors uncover a Persuasion Duality: models that reason explicitly are both better at persuading others and harder to persuade themselves. Weaker models give in more easily, subjective tasks make persuasion easier, and sharing “thinking content” boosts influence. At the same time, reasoning or step-by-step thinking helps models stay skeptical. The paper also explores how persuasion spreads across agents and proposes simple prompting method to make systems more robust.

**Strengths:**

•	It’s an interesting research topic - using different LLM model families as persuader and persuadee and showing the asymmetry is novel.

•	The scope of LLM selection is comprehensive and the comparison between thinking and thinking mode is interesting

•	The authors have considered both subjective and objective domains and analyzed separately

**Weaknesses:**

•	Section 1: Misinterpretation of the literature: “Initial research (Breum et al., 2024) on LLM persuasion has operated largely under the assumption that persuasive efficacy is a direct function of model scale.” -> Breum et al. (ICWSM 2024) didn’t make a “bigger model, more persuasive” claim, nor do they test model size. Their experiments use a single model (Llama-2-70B-chat) and focus on which persuasion strategies work.

•	Section 1: The terms “model’s internal cognitive architecture.” and “cognitive process,” are not  clearly defined. The term can be confused with “Cognitive Architectures for Language Agents (CoALA)” proposed in this well-known paper “Sumers, T., Yao, S., Narasimhan, K., & Griffiths, T. (2023). Cognitive architectures for language agents. Transactions on Machine Learning Research.”, which I suppose is what the authors were thinking about.

•	Section 2: The definitions of “Persuasive Metrics” are not clear. On one hand, you wrote “we define a prompt function f (q, c; LLM) that takes the question q, the contextual information c, to generate a responseˆ a.”, which seems that it’s the persuader’s persuasive message, but on the other hand, you wrote “ˆ ai = f (qi, ci; LLM) is the persuadee’s response.” In addition, it’s unclear what you meant by “contextual information c”? What does it include? I think you want to provide a concrete example to clarify each notation.

•	Section 2: The motivation and rationale for the metrics PR. RR, OR should be more clear.  Also, the notation “evaluation set S= qi, a0 i |(qi, ai) ∈Q∧a0 i = ai” should be better articulated.


•	Section 3.2: To be honest, the difference between Figure 1a and 1b seems small and only constrained to one or two columns. The first 5 columns, and column 6 and 7 seems identical across 1a and 1b. You should report statistical tests and effect size.

•	Section 3.2  Figure 1. The heatmap scales are different across 1a and 1b. I think you should use the same scale for fair comparison.


•	Figure 3 and Figure 4: You want to add the estimate for standard error/confidence interval. Especially for Figure 4, the variability between models seems large and the average line may not be meaningful.

•	Section 3.2, In general, all the claims should be backed up by more rigorous statistical testing given the presence of random noise.


•	Section 3.3.2, Figure 6. The difference in PR between “w/ thinking content” and “padding” is extremely close (65.8% vs. 62.3%). You should conduct statistical tests to make sure the difference is actually significant.

•	Section 3.3.2, Figure 6. In addition, if padding can already reach such a high PR (close to w/ thinking content), it undermines the value of thinking content.


•	Section 3.3.2. For the “Replace” condition,  why do you call it “flawed considering that it is still “valid” thinking content but just from a different LRM ”? Why does it have such low PR? You should be more clear on the Replace condition implementation. A better “Replace” condition would be a reasoning path from a totally unrelated conversation.

•	Section 3.3.2. For padding conditions, what if you simply repeat the persuasion statement in the “w/o Thinking condition” multiple times to the same length as w/ Thinking content? Maybe the reason that thinking content has higher PR has nothing to do with thinking process per se, but just due to the longer content that is aligned with the persuasion target stance.


•	Section  3.5. The result about multi-hop persuasion lacks depth in analysis and explanations.

•	Section 4.1. Figure 10. You should clarify which LLMs you are evaluating and how you derive the average attention score (which I suppose is based on the attention matrix of the first attention layer?)


•	Section 4.1. Figure 10.  Since there are multiple layers of attention, does it make sense to look at the attention score derived from the first attention layer alone? What if the tokens have low attention score in the first later, but end up contributing largely to higher attention layers? Please include some literature to justify your approach.

•	Conclusion section. As mentioned above, given the tiny gap PR between padding condition and thinking condition, I think the following statement is unfounded: “This paper challenges the scale-centric paradigm of persuasion in LLMs by empirically demonstrating that persuasive dynamics in MAS depend fundamentally on the underlying cognitive processes, particularly thinking process.”


•	There is one critical aspect the authors failed to consider: the model inherent tendency/stance. Previous studies [1] [2] show that LLM has “inherent stance” and it tends to converge/drift to that inherent bias with or without social interaction/debate. To fairly compare different LLM/LRM in terms of their persuaded-rate (PR), and remain-rare (RR), it is important to first know the baseline PR/RR in the absence of persuasion. If you simply prompt the LLM/LRM about the same statement, according to the literature, there should also be a non-zero drift in stance, which can contribute to non-zero PR and OR. You should acknowledge and address this in the paper.

**Questions:**

•	Section 3.1: You wrote “In the experiments, if the persuadee’s initial response is either support or oppose, the persuasion target is set to neutral. If the initial response is neutral, the persuasion target is randomly assigned to either support or oppose.”. Why didn’t you test the condition where the initial response is “support”, and the target is “oppose”, or vice versa?

•	Section 3.2  Figure 1. Why are you including both instruct (LLM) and thinking models (LRM) in the same matrix? I thought your claim is that they are fundamentally different. You should use some visual aid to differentiate these two types of models in the figure.

•	Section 3.4.2. Figure 8: The legend is confusing. What are you actually showing? For example, what does “Llama-3-8B vs. Qwen2.5-7B w/ cot” mean? Are you showing the gap between the two models?

---

### Official Review · Reviewer_7JBM · 2025-11-01

**Soundness:** 2
**Presentation:** 2
**Contribution:** 2
**Rating:** 4
**Confidence:** 4

**Summary:**

This paper investigates the role of internal reasoning processes in model to model persuasion interactions: both the degree to which the reasoning process in LRMs aids in persuasion of other models, and the resistance to persuasion afforded by the reasoning process. While parameter size is generally correlated with increases in persuasive ability, diminishing returns in the relationship between model size and persuasion leaves significant variance unexplained. The authors demonstrate the role of internal reasoning in both resistance to persuasion and effectiveness in persuading other models through analyses of multi-agent systems (MAS) in both subjective and objective tasks. Finer grained analysis of the effect of internal reasoning on persuasion and persuasiveness demonstrates the role of length and coherence of reasoning chains. Additionally, follow-up prompt-based interventions support the effectiveness of encouraging the model to critically evaluate unsupported claims in persuasion resistance.

**Strengths:**

The comparison of persuasion effectiveness across both objective and subjective benchmarks highlights an interesting interaction between model reasoning and parameter size depending on the task type, with implications for the decision to employ model reasoning in various downstream tasks. Additionally, the analysis of multi-hop persuasion chains is a novel contribution, and indicates interesting non-linear relationships in the accumulation of persuasion effects across models.

**Weaknesses:**

The prompt mitigation intervention supports the findings of the attention analysis, namely that high model attention scores for persuasive language affect persuasion, and that explicit instruction to avoid persuasive language mitigates this effect. However, the attention analysis
itself needs significant improvement to support the general claim that the model prioritizes persuasive information. There is no causal analysis to demonstrate the influence of these attention patterns on model outputs–what happens when attention to superficial tokens is ablated? Merely showing attention activations over a single prompt for a single layer is not sufficient to make a causal claim. Additionally, it is problematic that little to no information is provided on the case study itself. Which model’s internal representations are being evaluated? What layer of the model is this/ How are attention scores obtained? While the claim and following prompt-based intervention are compelling, I would appreciate a more comprehensive analysis of these attention patterns across prompts/domains/models.

**Questions:**

The authors demonstrate the effectiveness of prompt-level mitigation where llama-3-8b-instruct acts as the persuader, but how does this analysis extend to other models? While the results are consistent for the interaction of llama-3-8b-instruct with various persuadee models, it would be nice to see a consistent relationship in other model interactions.

---

### Note · Authors · 2025-11-21

I have read and agree with the venue's withdrawal policy on behalf of myself and my co-authors.